# An Automatic Bearing Fault Diagnosis Method Based on Characteristics Frequency Ratio

**DOI:** 10.3390/s20051519

**Published:** 2020-03-10

**Authors:** Dengyun Wu, Jianwen Wang, Hong Wang, Hongxing Liu, Lin Lai, Tian He, Tao Xie

**Affiliations:** 1State Key Laboratory of Robotics and System, Harbin Institute of Technology, Harbin 150001, China; wudyziyu@163.com; 2Science and Technology on Space Intelligent Control Laboratory, Beijing Key Laboratory of Long-life Technology of Precise Rotation and Transmission Mechanisms, Beijing Institute of Control Engineering, Beijing 100194, China; hongwang1981@163.com (H.W.); liuhongxingcn@163.com (H.L.); lailinabc@sina.com (L.L.); 3School of Transportation Science and Engineering, Beijing 100191, China; zy1813322@buaa.edu.cn (J.W.); hetian@buaa.edu.cn (T.H.)

**Keywords:** rolling bearings, automatic fault diagnosis, envelope analysis, characteristics frequency ratio, ensemble empirical mode decomposition

## Abstract

Bearing is a key component of satellite inertia actuators such as moment wheel assemblies (MWAs) and control moment gyros (CMGs), and its operating state is directly related to the performance and service life of satellites. However, because of the complexity of the vibration frequency components of satellite bearing assemblies and the small loading, normal running bearings normally present similar fault characteristics in long-term ground life experiments, which makes it difficult to judge the bearing fault status. This paper proposes an automatic fault diagnosis method for bearings based on a presented indicator called the characteristic frequency ratio. First, the vibration signals of various MWAs were picked up by the bearing vibration test. Then, the improved ensemble empirical mode decomposition (EEMD) method was introduced to demodulate the envelope of the bearing signals, and the fault characteristic frequencies of the vibration signals were acquired. Based on this, the characteristic frequency ratio for fault identification was defined, and a method for determining the threshold of fault judgment was further proposed. Finally, an automatic diagnosis process was proposed and verified by using different bearing fault data. The results show that the presented method is feasible and effective for automatic monitoring and diagnosis of bearing faults.

## 1. Introduction

Moment wheel assemblies (MWAs) and control moment gyros (CMGs) have been widely used in satellite attitude control and large angle slewing maneuvers over the years. High-speed rotating systems are supported by a pair of angular contact ball bearings with different sizes and capacities [1]. As reliability is of paramount importance, bearings with high precision class are selected. As one of the critical components, bearing failures lead to partial and total mission failure or performance degradation of the satellite [2,3]. Satellite in-orbit failure statistics show that a large part of satellite failures come from the attitude and orbit control sub-system, and more than half of these failures are caused by the bearings [4]. Therefore, long-life experiments of ball bearings on the ground become an essential and effective solution, and effective methods for the diagnosis of the operating conditions of bearings in MWAs are required [5].

For rolling bearings under highspeed and low loading conditions such as MWAs bearings, local scratching damage caused by skidding is a common failure [6]. When the bearing component’s surface is locally damaged, the high-frequency impact response between components during the operation will be excited [7,8] and the vibration signals exhibit an amplitude modulation phenomenon that combines the characteristic frequency of the bearing defect with the structural resonances [9]. Therefore, as a sensitive and effective method [10], vibration measurements are widely used to detect bearing defects in the fields of aviation [11], transportation [12], energy [13], and other fields. Among many diagnostic methods, the envelope detection (ED) method is one of the most commonly used and effective methods in vibration-based bearing fault diagnosis [14,15], which was presented by Mechanical Technology Inc. in the early 1970s [16] and was originally called the high frequency resonance technique [17].

So far, more attention has been paid to the research on bearing fault diagnosis. A review of envelope detection was presented by Randall et al. [18] and Tyagi et al. [19], and envelope detection is being continuously improved to diagnose weaker fault information under strong noise. Klausen et al. [20] presented a method for analyzing multiple narrow bands and bearing faults could be detected autonomously by a narrow-band envelope spectrum-based algorithm. The accelerated life test has verified the performance of the proposed method. Feng et al. [21] studied the performances of several envelope detection methods for extracting fault features with wireless sensor nodes, which showed that the spectral correlation and short-time RMS (root mean square) based methods can well reveal the simulated three types of bearing faults with a significantly improved computation speed. Tyagi et al. [19] aimed to address the problem of traditional envelope detection being highly sensitive to the envelope window, and employs a particle swarm optimization method to select the most optimum envelope window to band pass the vibration signals induced by fault rolling element bearings. In order to determine an informative spectral frequency band for generating an enhanced/squared envelope spectrum, Wang et al. [22] proposed a simple and fast guideline and conducted an experiment to highlight its superiority by comparing it with the fast Kurtogram. Tsao et al. [23] introduced empirical mode decomposition to select an appropriate resonant frequency band for characterizing the characteristic frequencies of bearing faults by using the envelope analysis subsequently, and the experimental results showed that the proposed method can diagnose the bearing fault types efficiently and correctly.

The methods above-mentioned are mainly aimed at the diagnosis of early bearing failure on the ground. However, for space bearings, certain fault characteristics can also be captured in normal bearings. The problem at this time is not to extract earlier fault characteristics, but to distinguish the characteristics between normal and fault bearings. Moreover, the envelope spectrum contains rich information, thus manual identification requires professional knowledge and experience of the diagnostic staff and a large workload. To solve these problems, based on the bearing vibration experimental data, this paper proposes a method for calculating the characteristic frequency ratio, which is used to quantitatively evaluate bearing failures, and automatic fault diagnosis and quantitative diagnosis processes are further proposed.

The rest of this paper is organized as follows. First, a vibration test of an MWA’s bearing assembly is carried out in Section 2, which provides the data for the envelope analysis. In Section 3, an improved envelope analysis method is presented based on the ensemble empirical mode decomposition (EEMD) method to acquire the fault features of the vibration signals. In Section 4, the method of the characteristic frequency ratio is put forward and in Section 5, the proposed method is verified by using different bearing fault data. Finally, the conclusions are drawn in Section 6.

## 2. Experimental Data Acquisition

Figure 1 shows the experimental device of the satellite flywheel bearing assembly. The assembly’s rotor is supported by a pair of angular contact ball bearings. The inner ring of the bearing is connected to the fixed shaft, and the rotating housing supported by the outer ring is connected to the drive motor. Therefore, the inner ring of the bearing is fixed and the outer ring rotates. The accelerometer was mounted on one end (top) of the fixed shaft. Due to the higher resonance frequencies of the components, a higher vibration signal sampling frequency of 25,600 Hz was used during testing. The normal bearings and various fault bearings that actually occur during long-term testing had be tested, and the data in various states were obtained to study the automatic diagnosis method. Some of the damage bearing elements are shown in Figure 2.

Figure 3a,b show the time-domain waveforms and their spectrum diagrams of normal bearings. It can be seen from Figure 3 that the vibration signal components of the normal bearing are relatively complicated.

For a stationary inner ring (*f_r_*) and rotating outer ring, the fundamental frequencies are derived from the space bearing geometry as follows:

Rolling element rotational frequency
(1)$$f_{b}=\left(\frac{D_{m}}{2D_{b}}\right)\left(1-\frac{{\left(D_{b}cos\alpha \right)}^{2}}{D_{m}^{2}}\right)f_{r}$$

Ball pass frequency of inner ring (BPFI)
(2)$$f_{ip}=\left(\frac{Z}{2}\right)\left(1-\frac{D_{b}cos\alpha }{D_{m}}\right)f_{r}$$

Ball pass frequency of outer ring (BPFO):
(3)$$f_{op}=\left(\frac{Z}{2}\right)\left(1+\frac{D_{b}cos\alpha }{D_{m}}\right)f_{r}$$

Fundamental train frequency relative to inner ring:
(4)$$f_{ic}=\left(\frac{1}{2}\right)\left(1-\frac{D_{b}cos\alpha }{D_{m}}\right)f_{r}$$

The characteristic frequencies of the defect bearing can be defined as the fundamental frequencies and their multiple frequencies.

According to the calculation formulas of the characteristic frequencies of bearing failures, the characteristic frequencies of the local damage of each component under 3000 rpm are shown in Table 1.

## 3. Fault Feature Extraction Method Based on Improved Envelope Detection method

### 3.1. The Proposal of the Method

In the vibration test, the vibration signal collected by the sensor contains not only the bearing fault information, but also a large amount of background noise. The ED technique involved band passing the vibration signal while keeping the central frequency of the pass band at one of the resonances, and its diagnostic performance will degrade if the selected central frequency and bandwidth of the bandpass filter is not optimum. The selection of the envelope window is a hot topic in the field of bearing fault diagnosis. Some adaptive screening methods such as the local mean decomposition (LMD) and the empirical mode decomposition (EMD) have been introduced to select frequency bands automatically, and have achieved good results in bearing diagnosis [23,24]. In order to overcome the mode mixing of EMD, the ensemble empirical mode decomposition (EEMD) was presented and can extract the weak impact component more effectively [25]. Here, the improved EEMD method was introduced to select appropriate IMFs for the envelope analysis of bearing fault diagnosis. At present, the EEMD-based bearing diagnosis method mainly focuses on the diagnosis of a single fault such as the pitting or spalling of the ground bearing component. Therefore, a few studies have selected the most sensitive IMF to extract fault features according to the kurtosis of the IMF, and have achieved good results.

For MWA bearings, local scratch damage due to skidding is a common failure. As a result, the raceways and balls are often scratched against each other at the same time, which often results in local failures. Moreover, it usually presents a coupled fault. At the beginning, because the damage is minor and is a multi-faults form, not only the denoising method but also the demodulation method with less loss of fault information is needed. Therefore, in the case of a coupled fault, the IMFs that respond to the faults should be fully selected for further processing. Therefore, a multi-IMFs selection method should be proposed based on a new standard. Here, an improved EEMD method based on kurtosis for IMF selection is introduced to extract the fault features, which is described below.

For an IMF component, let the *i*th time series point be *x_i_*. Then, the kurtosis *K* of the IMF to be defined as
(5)$$K=\frac{1}{N}\overset{N}{\underset{i=1}{\sum }}{\left[\frac{x_{i}-\mu }{\sigma }\right]}^{4}$$
where *N* is the length of the time series point; *μ* is the mean value; and *σ* is the standard deviation of the IMF.

If the probability density function of the IMF satisfies the normal distribution, the kurtosis *K* = 3 regardless of the variance. If the IMF component contains the shock signal caused by the damage, the shock component will increase the probability density of the vibration signal with a larger amplitude, and the corresponding *K* will be greater than three. Therefore, any IMF with a kurtosis greater than three may contain fault information. Moreover, it is possible that different faults result in different IMF components. Therefore, it is possible to select all IMFs with a *K* greater than three for further processing without losing fault information [20]. In this paper, IMFs with a *K* greater than three are added together to reconstruct a new signal, and the signal is then used to extract the fault feature frequencies based on envelope analysis. Thus, a coupling fault diagnosis method for light-load aerospace rolling bearings was introduced based on EEMD and kurtosis and be described as follows:

(1) The vibration signal of the bearing assembly was obtained by the experimental device. It is worth pointing out that the sampling frequency of the bearing assembly vibration needs to meet a certain oversampling rate. Given that the resonance frequencies of the bearing components are generally high, 25,600 Hz was used in this study.

(2) Then, the EEMD was used to decompose the test signal into a set of IMF components. The signal to be analyzed needs a certain length of time. In this paper, signals with a length of 1 s and about 50 cycles were selected.

(3) The kurtosis for each IMF component was calculated. All IMFs with a kurtosis greater than three were added to reconstruct a new signal.

(4) The Hilbert transform was used to process the reconstructed signal to obtain its envelope signal, and the envelope spectrum of the bearing failure was calculated from the envelope signal by Fourier transform.

(5) The envelope spectrum was analyzed and the fault type of the rolling bearing according to the characteristic frequency of the fault was estimated.

### 3.2. The Verification of the Method

For MWA bearings, local scratching damage caused by skidding is a common failure. Therefore, the raceway and the ball are often stabbed at the same time, so it is a coupling fault. Here, we took the scratching damage as the diagnosis object.

Figure 4 shows the fault feature extracting results of the ball–inner ring scratching. According to Table 1, when the rotational speed of the rolling bearing was 3000 rpm, the theoretical fault characteristic frequency of the inner ring was 356.1 Hz while that of the ball was 124.7 Hz. In Figure 4, the envelope spectrum of a single inner ring fault signal contains the fault characteristic frequency of the bearing inner ring (357 Hz, very close to the theoretical value of 356.1 Hz) and its second harmonic frequency (751 Hz). The ball characteristic frequencies (126 Hz and 251 Hz) can also be seen in Figure 4.

The fault feature extracting results of the ball–outer raceway scratching is shown in Figure 5. The theoretical fault characteristic frequency of the outer ring was 243.9 Hz while that of the ball was 124.7 Hz. In Figure 5, the envelope spectrum contains the fault characteristic frequency of the bearing’s outer ring (245 Hz, very close to the theoretical value of 243.9 Hz) and its second harmonic frequency (490 Hz). The ball characteristic frequencies (251 Hz, nearly twice of the characteristic frequency) can also be detected in Figure 5.

In order to verify the adaptability of the method, the scratched ball was replaced by a normal ball. The measured vibration signal and the envelope spectrum are shown in Figure 6. In Figure 6, the envelope spectrum contains only the fault characteristic frequency of the outer ring and its multiple frequencies (246 Hz, 491 Hz and 735 Hz).

Based on the points discussed above, the improved EEMD can diagnose the faults accurately regardless of whether it is a single fault or coupling fault.

Aiming at the vibration signals of normal bearings as shown in Figure 3, the results processed by the improved EEMD method are shown in Figure 7. It can be seen that although the larger frequency in the envelope spectrum is not the characteristic frequency, some frequency components closely related to the characteristic frequency of the bearing are still visible. This means that the bearing condition needs to be judged based on the expert’s experience, which brings difficulties to the real-time health monitoring of space bearings for long life. Therefore, a method for distinguishing between a faulty and a normal bearing is needed, and in particular, a method capable of automatic diagnosis.

## 4. The Automatic Fault Diagnosis Method Based on Characteristic Frequency Ratio

Based on the conclusions in Section 3, it is evident that the fault feature information extracted by the improved EEMD method is clear and accurate, which can diagnose the coupled faults effectively at different speeds. However, in this method, the fault characteristic frequencies still have to be detected in the envelope spectrum by the use of the method, which is limited by the experience and knowledge of testers to some extent in practical applications.

It can also be concluded from Section 3 that the amplitude or energy of the characteristic frequencies in the envelope spectrum is relatively high under fault conditions, thus the characteristic frequency ratio is proposed in this paper, which can be used to determine the fault automatically by calculating the amplitude index of the characteristic frequency of the envelope spectrum.

### 4.1. The Characteristic Frequency Ratio

(6)$$A_{ea}=\frac{1}{N_{e}}\overset{N_{e}-1}{\underset{i=0}{\sum }}A_{e}\left(f_{i}\right)$$
where $A_{\mathrm{ea}}$ is the average value of the spectrum, and $N_{e}$ is the number of spectral lines of the analysis bandwidth in the envelope spectrum.

During the actual operation of the bearing assembly, due to the rotating speed instability caused by skidding fault and the error between the calculated parameters and the actual parameters, the calculated fault characteristic frequencies usually deviate from the corresponding actual characteristic frequencies. Hence, $A_{\mathrm{ed}}\;$is calculated as
(7)$$A_{ed}=\frac{1}{n_{e}}\overset{n_{e}}{\underset{i=1}{\sum }}max\left[A_{e}\left(if_{d}-\Delta f\right),A_{e}\left(if_{d}-\Delta f+f_{ss}\right),\cdots ,A_{e}\left(if_{d}\right),\cdots ,A_{e}\left(if_{d}+\Delta f-f_{ss}\right),A_{e}\left(if_{d}+\Delta f\right)\right]$$
where $f_{\mathrm{ss}}$ is the frequency resolution and Δf is the frequency interval for searching characteristic frequencies.

Considering the deviation of $f_{d}$ from the theoretical value and the difference between the characteristic frequencies, Δf was set as 2 Hz.

Then, the characteristic frequency ratio δA can be defined as
(8)$$\delta A=\frac{A_{ed}}{A_{ea}}$$
The value of δA can be used to judge whether there is a fault and the severity of the fault.

### 4.2. Automatic Bearing Fault Diagnosis Process Based on δA

The specific calculation process of the automatic bearing fault diagnosis method based on the characteristic frequency ratio is shown in Figure 8.

(1) Signal decomposition and reconstruction. Decompose and reconstruct the bearing vibration signals based on the improved EEMD method proposed in Section 3.

(2) Characteristic frequency extraction. Resonant demodulate the reconstructed signal to obtain the envelope spectrum.

(3) Calculate the characteristic frequency ratio of the envelope spectrum according to Equations (6)–(8);

(4) Automatic diagnosis: Comparing the obtained results with threshold $d_{\mathrm{limit}}$ and observing whether the various characteristic frequency ratios exceed the limits and the extent of the limits to determine whether the bearing element has failed.

## 5. Determination of Automatic Discrimination Threshold

The fault discrimination threshold of the characteristic frequency ratio is very crucial for the automatic fault diagnosis of bearings. Theoretically, δA=0 means that there is no fault characteristic line in the envelope spectrum, which indicates no damage in the bearing. However, according to the analysis results in Section 3, the characteristic frequency line of a normal bearing is generally not zero. In light of this situation, the actual bearing vibration data were utilized in this paper to establish the discrimination threshold of the characteristic frequency ratio. According to the diagnosis process, the characteristic frequency ratio is calculated for normal bearing data, outer ring fault, ball-inner ring scratching fault, and ball-outer ring scratching fault, which is prepared to find the determination method of the discrimination threshold.

In order to have practical significance, 30 samples were used for each condition, and the envelope spectrum ratios were calculated. The average value of the obtained characteristic frequency ratios is shown in Table 2. It can be seen from Table 2 that the ratios of the normal bearings were small, and the average value was less than 2. When a bearing fault occurs, the characteristic frequency ratio of the corresponding fault component increases significantly, and the smallest reached 3.573 (corresponding to the ball in the ball–outer ring scratching fault). Another phenomenon is that when there is a fault, the characteristic frequency ratio corresponding to the non-faulty element will also rise, for example, the ball ratio in the outer ring fault condition increased to 2.438. In terms of the distribution of the averages of the ratios, it seems easy to determine whether there is a fault. However, considering the dispersion of the data, the ratios were analyzed based on the box-plot method, as shown in Figure 9.

It can be seen from Figure 9 that it is intuitive and effective to distinguish normal and fault bearings according to the characteristic frequency ratio. For normal bearing data, various fault characteristic frequency ratios are low, in contrast, the fault characteristic frequency ratio will be significantly high. For the ball–outer ring scratching, the increase of the ball characteristic frequency ratio is not very significant; while under the outer ring failure, the ratio rises significantly, resulting in the lower limit of the ratio in the ball–outer ring scratching fault being higher than the one in the outer ring fault. According to Figure 9, the upper boundary without fault was 3.48 and the lower boundary with the fault was 2.88. Data between the two boundaries will result in misjudgment. Statistics showed that the two conditions of normal and ball–inner ring coupling were completely correct. There were six groups of data recognition errors in the outer ring fault with an accuracy rate of 93.3% (84/90). There were 16 sets of data errors during the ball–outer ring coupling recognition, with an accuracy rate of 82.2% (74/90). Overall, the accuracy rate was 93.89% (338/360).

Therefore, it is difficult to determine the threshold for fault diagnosis effectively, and false alarms or missed alarms are inevitable. As such, the method needs to be improved in order to enhance the recognition effect.

In order to enhance the discrimination effect, it is necessary to solve the problem where a characteristic frequency ratio corresponding to non-faulty components will also rise. Here, a processing method is proposed: Take the minimum value of ratios (inner ring, outer ring, and ball) of each set of data as the reference denominator, and all the characteristic frequency ratios are divided by this value to obtain a new ratio result. In this way, the rise shift effect of the ratios corresponding to non-faulty elements is corrected to a certain extent. The data is re-settled according to this process and the resulting box-plot is shown in Figure 10. It can be seen from Figure 10 that the lower bounds of the ratios of all faults were above the upper bounds of the characteristic frequency ratios of all non-faulted elements, so the judgment threshold can be well determined at this time. The threshold can be determined as needed such as the maximum quartile of the non-faulty element (1.31), the maximum upper bound of the non-faulty components (1.53), the minimum quartile of the fault element (1.98), and the minimum lower bound of the fault element (1.58). In this paper, the threshold was determined to be 1.31, which was the largest quartile of non-fault elements. In all fault situations, the accuracy of fault identification using this method reached 100%. In this way, the characteristic frequency ratio can be used to diagnose bearing faults automatically and locate faulty components.

## 6. Conclusions

This paper introduced an improved EEMD envelope detection method and characteristic frequency ratio for automatic bearing fault diagnosis. By comparing the diagnosis results under different fault conditions, the following conclusions can be obtained:

Damage due to the scratching between two components of space bearings with light loading is commonly a coupling fault. The improved EEMD method can carry out the diagnosis of single and coupled faults effectively without the loss of fault information. However, certain characteristic frequency components can also be detected in normal bearings with light loading.

The proposed characteristic frequency ratio method is intuitive and effective to distinguish normal and fault bearings. For normal bearing data, various fault characteristic frequency ratios are lower, while for fault bearing data, the fault element’s characteristic frequency ratio will be significantly larger. However, the characteristic frequency ratio of non-faulty elements will also increase. Although the increase is much smaller than that of faulty components, it sometimes overlaps with the ratio of weak faults.

When the minimum ratio is used to weight all ratios, the corrected characteristic frequency ratios of non-fault components will be suppressed. At this time, the lower boundary of all the fault elements’ ratios are above the upper boundary of all the characteristic frequency ratios of the non-faulty elements. Thus, the fault can be distinguished from the non-fault effectively.

The method proposed in this paper can be used to realize the automatic identification of bearing faults, and is practical for different types of faults. It is suitable for monitoring bearings in long-term tests and can also provide references for other diagnosis fields using the envelope spectrum.

## Figures and Tables

**Figure 1 sensors-20-01519-f001:**
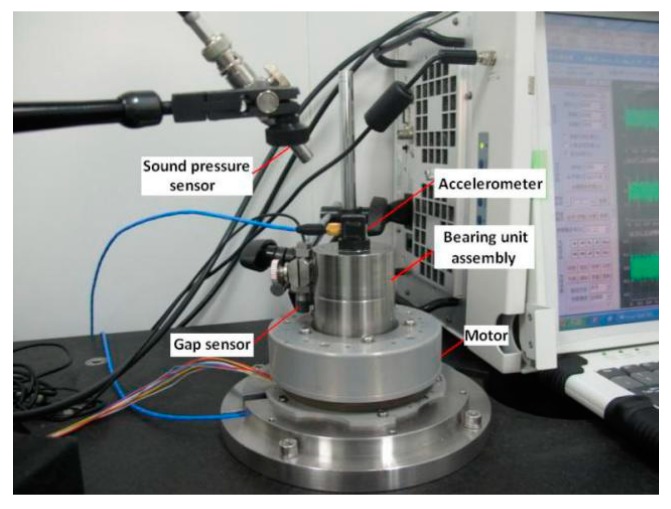
The experimental device.

**Figure 2 sensors-20-01519-f002:**
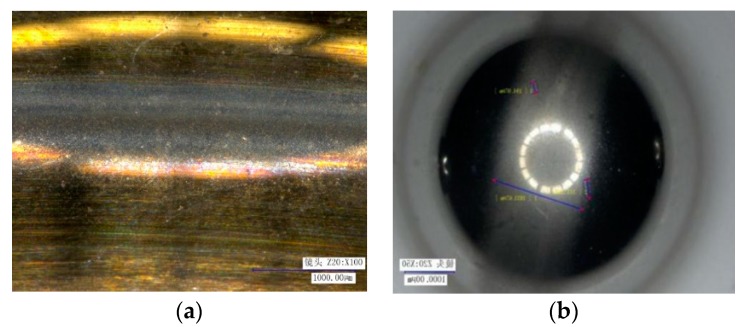
Damage elements of bearings. (**a**) The scratched inner ring. (**b**) The scratched ball.

**Figure 3 sensors-20-01519-f003:**
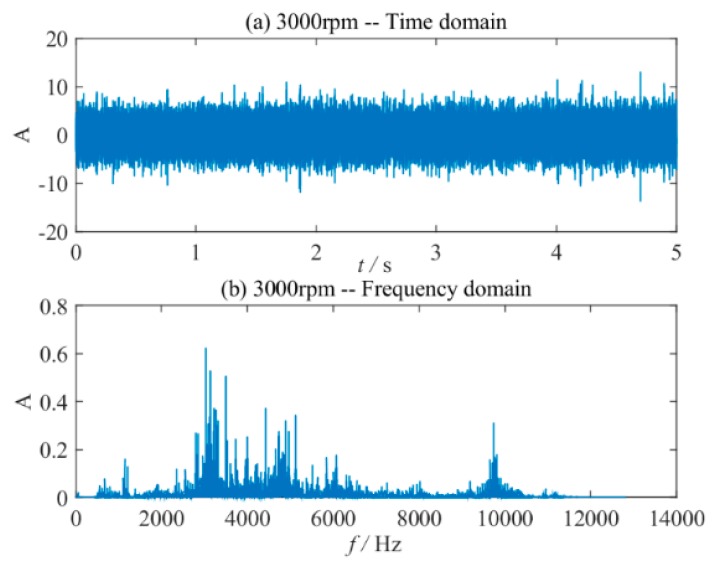
Normal bearing vibration signal. (**a**) The time-domain waveform at 3000 rpm; (**b**) the frequency spectrum at 3000 rpm.

**Figure 4 sensors-20-01519-f004:**
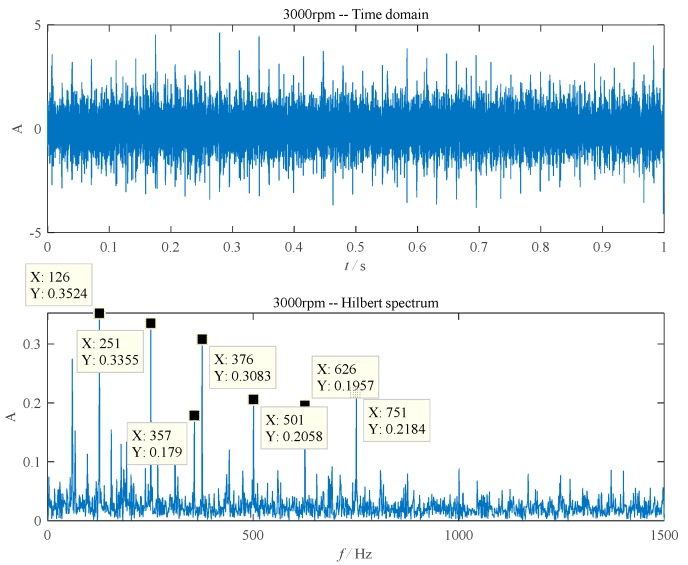
The ball–inner ring scratching fault. (**a**) The time domain waveform; (**b**) envelope detection based on improved EEMD.

**Figure 5 sensors-20-01519-f005:**
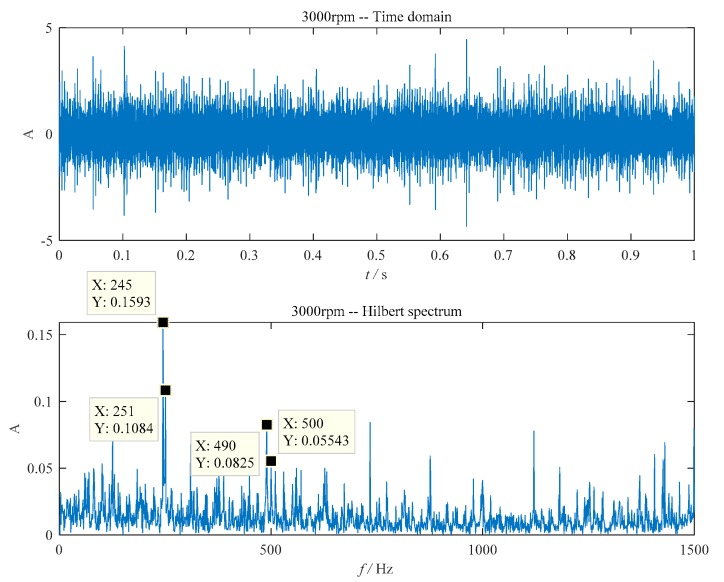
The ball–outer raceway scratching fault. (**a**) The time domain waveform; (**b**) envelope detection based on improved EEMD.

**Figure 6 sensors-20-01519-f006:**
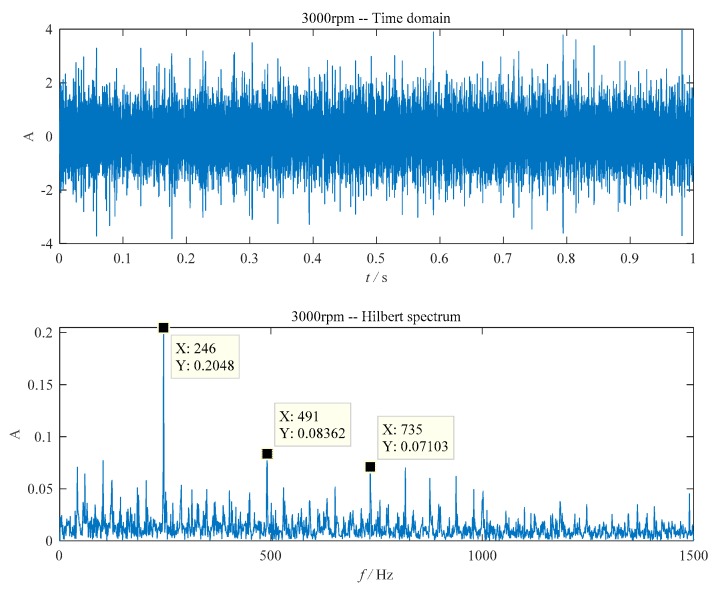
The Hilbert envelope spectrum. (**a**) The time domain waveform; (**b**) envelope detection based on improved EEMD.

**Figure 7 sensors-20-01519-f007:**
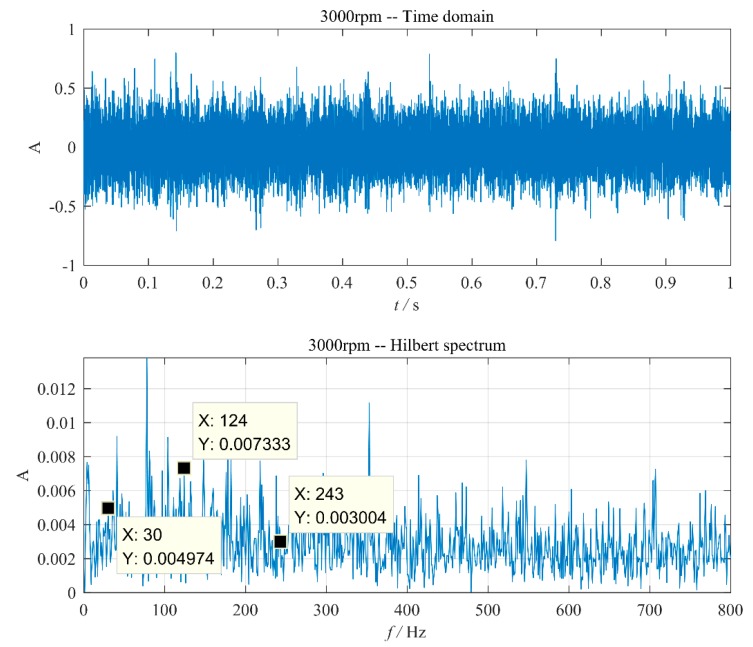
The normal bearing vibration signal. (**a**) The time domain waveform; (**b**) envelope detection based on improved EEMD.

**Figure 8 sensors-20-01519-f008:**
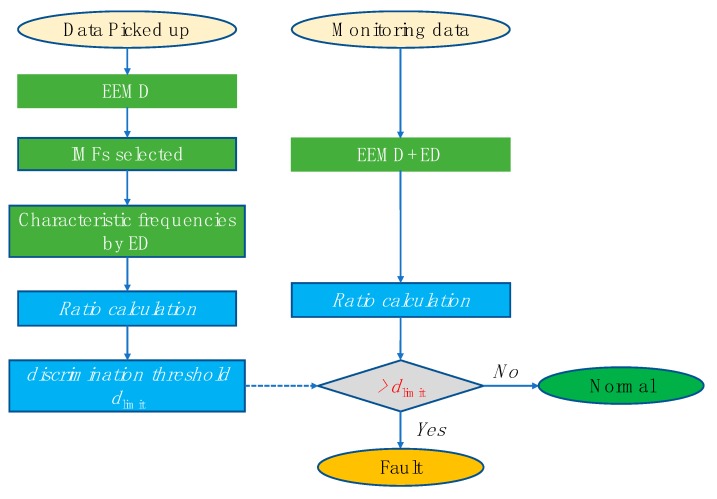
The automatic bearing fault diagnosis process.

**Figure 9 sensors-20-01519-f009:**
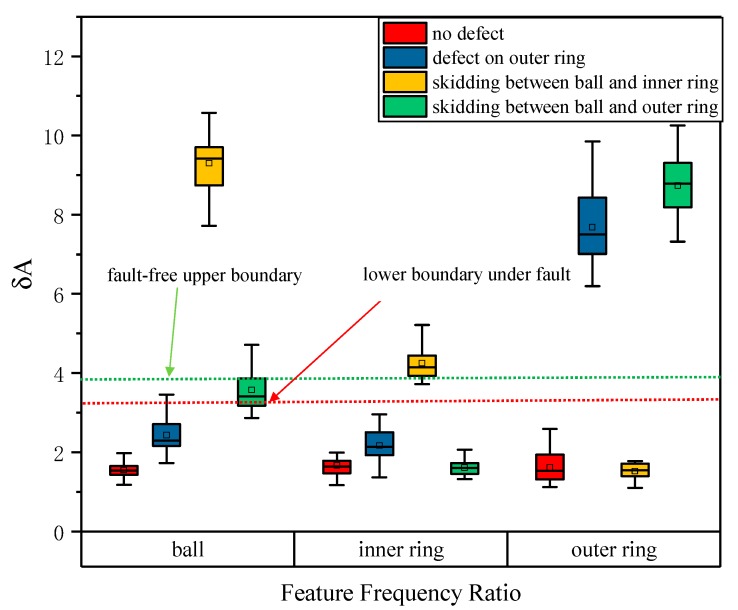
Box-plot diagram of the characteristic frequency ratio of various faults.

**Figure 10 sensors-20-01519-f010:**
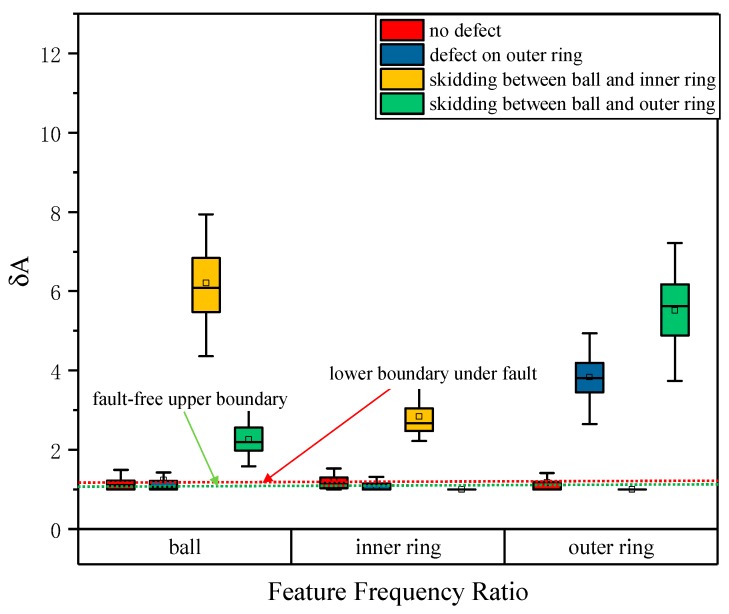
Box-plot diagram of the corrected characteristic frequency ratios of various faults.

**Table 1 sensors-20-01519-t001:** Characteristic frequencies of the local damage of bearing elements at different speeds.

	Rotating Speed	3000 rpm
Fault Elements		
$\;f_{ip}\;$		356.1 Hz
$\;f_{op}\;$		243.9 Hz
$\;f_{b}\;$		124.7 Hz
$\;f_{ic}\;$		29.7 Hz

**Table 2 sensors-20-01519-t002:** The characteristic frequency ratios of bearings in various states (average value).

	Elements	Inner ring	Outer ring	Ball
Fault Types				
Normal		1.667	1.617	1.553
Outer ring		2.173	7.682	2.438
ball-inner ring scratching		4.251	1.526	9.302
ball-outer ring scratching		1.608	8.735	3.573

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
