# Peer review of "An Automatic Bearing Fault Diagnosis Method Based on Characteristics Frequency Ratio"

_sensors, 2020, doi:10.3390/s20051519_

Round 1

Reviewer 1 Report

In this paper, the improved ensemble empirical mode decomposition is used for detecting faults in bearings. This work is well organized and the method sounds correct. I think this work could be accepted if the following concerns are properly considered.

1) As shown in Fig. 1, the sound pressure sensor is mounted in the device. Is this signal/variable used in your method?

2) The state-of-the-art of this research should be improved because some important references, such as A review of fault detection and diagnosis for the traction system in high-speed trains, may be supplemented in the revision.

3) In my opinion, the detection should be based on some knowledge of faults beforehand. In practice, these prior or knowledge can be obtained as expected?

Author Response

1) As shown in Fig. 1, the sound pressure sensor is mounted in the device. Is this signal/variable used in your method?

Response:During the test, both the vibration signals and the acoustic signals were picked up. However, the study found that acoustic signals are susceptible to environmental interference, and they contained less fault information than vibration signals. Therefore, this paper chose vibration signals for fault diagnosis instead of using acoustic signals.

2) The state-of-the-art of this research should be improved because some important references, such as A review of fault detection and diagnosis for the traction system in high-speed trains, may be supplemented in the revision.

Response:Based on the reviewer's suggestions, we have added three articles as follows:

  1. Ai Y. T., Guan J. Y.e, Fei C. W., Tian J., Zhang F. L., Fusion information entropy method of rolling bearing fault diagnosis based on n-dimensional characteristic parameter distance, Mechanical Systems and Signal Processing 2017, 88, 123–136.
  2. Chen H.T., Jiang B., A review of fault detection and diagnosis for the traction system in high-speed trains, IEEE Transactions on Intelligent Transportation Systems 2020, 21, 450–465.
  3. Miao Y. H., Zhao M., Lin J., Periodicity-impulsiveness spectrum based on singular value negentropy and its application for identification of optimal frequency band, IEEE Transactions on Intelligent Transportation Systems 2019, 66, 3127-3138.

3) In my opinion, the detection should be based on some knowledge of faults beforehand. In practice, these prior or knowledge can be obtained as expected?

Response:The basis of the method in this paper is to obtain the characteristic frequencies by using envelope demodulation. The characteristic frequencies are priori and can be obtained by calculating formulas (1) ~ (4). Since the parameters and operation speed of the bearing are known, the prior knowledge can be obtained.

Special thanks to you for your good comments.

Reviewer 2 Report

This paper presents a procedure for an automatic bearing fault diagnosis using the characteristics frequency ratio.

The authors defended their arguments regarding signal processing very well.

The different choices made are justified. The analysis made seems correct.

However, to better highlight the relevance of their analysis, the authors are invited to consider the following points:

  1. In the flowchart of Figure 8, the choice to make regarding the final test must be clarified, i.e. it makes sense to add yes and no on the arrows leading to “normal” and “fault”.
  2. The relevance of this analysis must be judged by the recognition rate for the different faults. Isn't it wise to use a test signal set? This observation concerns the decision taken from the organization chart (Figure 8) (i.e. the procedure described between lines 250-258) as well as the analysis presented in Figures 9 and 10. I suggest here to take a test set and compute the recognition rate for each fault: which will give us the efficiency of the proposed approach. A comparison with similar works is also welcome.

Some other minor remarks:

  1. The sentence between lines 6 and 9 of the abstract is long and confusing.
  2. In the last paragraph of the introduction, it is better to replace part by section.
  3. In formulas (1)-(4), there is no need to use * for the product, a space is enough.

Author Response

1. In the flowchart of Figure 8, the choice to make regarding the final test must be clarified, i.e. it makes sense to add yes and no on the arrows leading to “normal” and “fault”.

Response:Thanks to the reviewers for their suggestions, Figure 8 is modified.

2. The relevance of this analysis must be judged by the recognition rate for the different faults. Isn't it wise to use a test signal set? This observation concerns the decision taken from the organization chart (Figure 8) (i.e. the procedure described between lines 250-258) as well as the analysis presented in Figures 9 and 10. I suggest here to take a test set and compute the recognition rate for each fault: which will give us the efficiency of the proposed approach. A comparison with similar works is also welcome.

Response:This article tested 4 flywheel bearing assemblies. The data used to study each type of fault comes from different flywheel tests, not one test data. Therefore, the diagnosis results in this article are representative.

In addition, the recognition rate has been increased in the revised draft. details as follows:

“Statistics show that the two conditions of normal and ball-inner ring coupling are completely correct. There were 6 groups of data recognition errors in the outer ring fault, with an accuracy rate of 93.3% (84/90). There were 16 sets of data errors during the ball-outer ring coupling recognition, with an accuracy rate of 82.2% (74/90). Overall, the accuracy rate is 93.89% (338/360). (Page 10, Section 5. Determination of automatic discrimination threshold, Line 289-294)”

“In all fault situations, the accuracy of fault identification using this method reaches 100%.” (Page 10, Section 5. Determination of automatic discrimination threshold, Line 313-314)”

Some other minor remarks:

1. The sentence between lines 6 and 9 of the abstract is long and confusing.

     Response:The abstract was revised according the comment.

2. In the last paragraph of the introduction, it is better to replace part by section.

      Response:It is done.

3.In formulas (1)-(4), there is no need to use * for the product, a space is enough.

     Response:It is done.

Special thanks to you for your good comments.

Reviewer 3 Report

The article deals with bearing fault diagnosis method based on characteristics frequency ration. The article is clear and logically structured. There are a few inconsistencies, e.g. formulas 1-4 do not fully correspond to Table 1 (different names); some images have low resolution. The authors should also consider whether it would be appropriate to compare the results of their method with other existing methods (on the same experiment data).

Author Response

The article deals with bearing fault diagnosis method based on characteristics frequency ration. The article is clear and logically structured. There are a few inconsistencies, e.g. formulas 1-4 do not fully correspond to Table 1 (different names); some images have low resolution. The authors should also consider whether it would be appropriate to compare the results of their method with other existing methods (on the same experiment data).

Response:Thanks to the reviewer for the suggestions. Table 1 was modified according to formulas 1-4. Figure 8 was revised.

 The innovative point of this paper is to propose the characteristic frequency ratio based on the envelope diagnosis, and establish an automatic diagnosis method. At present, there is no relevant literature on the automatic diagnosis method of flywheel bearing faults. In addition, most of the current literature on bearing fault diagnosis focuses on the extraction of fault features rather than automatic identification techniques (except intelligent methods). Due to the limited amount of data in this paper, it is difficult to compare with the intelligent learning methods. More importantly, based on the existing experimental data, the method proposed in this paper has been able to achieve 100% accurate fault identification. Therefore, this article does not compare with other existing methods.

Special thanks to you for your good comments.